# Cancer and Non-Cancer Risk Concerns from Metals in Electronic Cigarette Liquids and Aerosols

**DOI:** 10.3390/ijerph17062146

**Published:** 2020-03-24

**Authors:** Jefferson Fowles, Tracy Barreau, Nerissa Wu

**Affiliations:** Environmental Health Investigations Branch, California Department of Public Health, Richmond, CA 94804, USA; tracy.barreau@cdph.ca.gov (T.B.); nerissa.wu@cdph.ca.gov (N.W.)

**Keywords:** e-cigarettes, vaping, heavy metals, cancer risk assessment, chromium, nickel, nanoparticles, pulmonary toxicity

## Abstract

We evaluated metal concentrations in e-liquids and e-aerosols from eight studies and estimated the range of corresponding cancer and non-cancer risks. Chromium and nickel were the leading contributors to cancer risk, with minor contributions from cadmium, lead, and arsenic. The increased cancer risks, assuming exposure to 2 mL/day, ranged from 5.7 to 30,000 additional cancers in a million e-cigarette users. The average cancer risk was 3 in 1000. Cancer risks in the mid to upper end of these ranges exceed acceptable levels. The hazard quotient (HQ) approach was used to evaluate non-cancer risks. Hazard quotients exceeding 1.0 indicate the possibility for non-cancer adverse health effects. Estimated exposures at the maximum reported concentrations of nickel, chromium, and manganese resulted in HQ values of 161, 1.1, and 1.0, respectively, with additional contributions from lead. The average concentration of nickel resulted in an HQ value of 14. We conclude from these studies that exposure to metals in e-cigarette liquids and aerosols may pose a significant cancer and non-cancer health risk at the mid and upper end of the reported ranges. The device design and heating elements appear to be the main source of metals in e-aerosols. The large range of metals within and across e-cigarette brands indicate the need for improvements in product design, enforced product safety regulations and manufacturing quality control. Implementation of such measures could reduce metal exposure in e-cigarette users.

## 1. Introduction

E-cigarettes are widely used nationwide [1,2]. The devices come in a variety of designs, with a wide range of possible flavoring agents, carrier solvents, and often nicotine or cannabinoids. In the case of nicotine e-cigarettes, a solvent “carrier liquid” such as glycerin or propylene glycol delivers the flavorings and nicotine. All of these devices employ batteries and metallic heating elements of varying design to aerosolize the liquids. The composition of the metal filaments is not standardized and varies substantially. It is well documented that the heating of e-liquids to temperatures exceeding approximately 200 °C using these metallic filaments results in a predictable formation of carbonyl compounds, including acetaldehyde, acrolein, and formaldehyde from the starting material glycols [3]. A number of published studies showed that e-cigarette aerosols, e-liquids, and their chemical flavorings can cause respiratory irritation, inflammation, and toxicity to lung and immune cells [4,5,6,7,8,9]. The inflammatory and irritant effects of e-cigarette aerosols in vitro appear to be independent of the presence or absence of nicotine, suggesting that non-nicotine components are responsible [4,10]. Studies have found a positive association between e-cigarette use and incidence of asthma among high school students, after adjusting for conventional cigarette use [11,12]. Disruption of lung lipid homeostasis and innate immunity by e-cigarette glycol carrier liquids was also reported in one study [13]. In 2019, an outbreak of acute lung injuries associated with vaping viscous oily liquids used in cannabis vaping products resulted in thousands of hospitalizations and numerous fatalities nationwide—the chemical causes of which are currently unknown [14]. This current paper focuses on longer-term health effects, such as cancer or chronic respiratory diseases, from the use of these devices in the delivery of nicotine. However, the exposure to metals is expected to apply, in principle, to vaping any heated liquid. Since the use of e-cigarettes is a relatively new phenomenon, there may not yet have been sufficient time to see these longer-term effects.

Metals that comprise the heating elements and tanks of e-cigarette devices have the potential to be released into the liquids or the aerosols during use. Metal contaminants in e-liquids, or e-cigarette aerosols, have been quantified in a number of recent studies and include lead, cadmium, chromium, cobalt, arsenic, antimony, manganese, tin, nickel, zinc, copper, aluminum, iron, tungsten, and barium [15,16,17,18,19,20,21,22,23,24]. A recent case report of an e-cigarette user with giant cell interstitial pneumonia concluded that the presence of cobalt in liquid from the device was consistent with a ‘hard metal’ lung injury [25]. Some of these metals (e.g., nickel and chromium) are known respiratory irritants and/or allergens.

This paper summarizes data on metal contents of e-cigarette liquids and aerosols and characterizes the range of exposures and health risks to users of these products (not bystanders), using published cancer potency factors and non-cancer reference exposure concentrations (RfCs) from the U.S. Environmental Protection Agency (USEPA) [26] and reference exposure levels (RELs) from the California Environmental Protection Agency (CalEPA) [27,28].

## 2. Materials and Methods 

A literature review, using the PubMed and Toxnet (National Library of Medicine) online public access databases, found eight unique studies, including a review article, that reported quantitative metal concentrations for e-cigarette aerosols or liquids from various e-cigarette devices (Table 1). These studies represented the only published reports that provided quantitative metal data in liquids or aerosols of these devices. These data sets were combined to describe the range and average of reported values used in our analyses.

A point estimate of 2 mL/day, which falls in the range of reported e-liquid consumed per day (1–10 mL/day), was used to calculate dose estimate [29]. For aerosol data, the exposure assessment was based on the metal concentration in 10 puffs and the average of 163 puffs/day published by Dautzenberg and colleagues [30]. Dose estimates were based on USEPA default exposure assumptions including a body weight of 70 kg and a daily inhalation volume of 20 m^3^/day [31], over 70 years. Absorption was assumed to be 100% of the inhaled material. Cancer potency factors and non-cancer reference values [26,27,28] were used to estimate the increased cancer and non-cancer risk associated with each metal (Table 2). We used the National Ambient Air Quality Standard (NAAQS) and CalEPA’s Maximum Allowed Daily Limit (MADL) for lead (Pb) in our quantitative non-cancer risk calculations since RfC and REL values are not available [32,33]. The valence state of chromium was conservatively assumed to be in the VI+ state for all chromium e-liquid measurements, due to lack of metal speciation data. The implications of this assumption were considered.

Cancer risk estimates were based on the following equation (Equation (1)):Cancer risk = Cancer potency (μg/kg/d)^−1^ (see Table 2) × daily dose (μg/kg/d)(1)

Cancer risks exceeding 1 × 10^−5^ (1 in 100,000 extra cancers in the population) were considered significant.

Chronic, non-cancer risks were estimated using a hazard index (HI) approach (Equation (2)), where
Non-cancer hazard quotient (HQ) = Daily dose (μg/kg/d)/(REL or RfC) (μg/kg/day)(2)

The HI is the sum of the HQs based on the target organ. HI exceeding 1.0 indicates a potential for non-cancer adverse effects.

Air concentration values were converted to external doses using an assumed inhalation rate of 20 m^3^/day and a default 70 kg body weight.

## 3. Results

The summed contributions of each metal for cancer or combined non-cancer risk at low, high, and average metal concentrations are shown in Table 3 and Table 4. Based on the average reported nickel and chromium concentrations, and using CalEPA cancer potency values, cancer risks were estimated at 5.2 × 10^−5^ and 3.1 × 10^−3^ respectively; at the high end of the reported range, cancer risks were estimated at 5.9 × 10^−4^ for nickel and 3.1 × 10^−2^ for chromium. Using USEPA cancer potencies, nickel and chromium present risks of 9.7 × 10^−5^ and 2.6 × 10^−4^ at the average values, respectively, and 1.1 × 10^−3^ and 2.5 × 10^−3^ at the highest reported concentrations. Cancer risks exceeding 1 × 10^−4^ are generally considered unacceptable by the USEPA [34]. Even at the lowest reported concentration, the cancer risk from chromium (5.7 × 10^−6^) is not negligible. Cadmium exposures at the average and high end of the range, results in cancer risks of 1.2 × 10^−5^ and 5.2 × 10^−5^, respectively.

Non-cancer HQ values were summed for each metal to derive the HI; respiratory and nervous system effects are shown separately in Table 5. A HI > 1.0 signals the potential for adverse health effects. There is no RfC for nickel, and so it was not included in the estimated HI value in the USEPA based calculations. Using CalEPA RELs, and based on the high end of the range for nickel levels, a HI of 162 was estimated for respiratory effects. Nickel accounted for most of the risk, with an HQ of 161. Manganese contributed an additional HQ of 1.0. Using average values, a HI of 14.5 was calculated based on respiratory effects, with nickel being the primary contributor (Table 5).

The HQ for chromium exceeds 1.0 at the high end of the range using CalEPA or USEPA reference values (Table 3 and Table 4). The HQs in Table 4 were estimated using a USEPA RfC (0.1 μg/m^3^) based on the particulate form for chromium, which is not the most conservative reference value available. If the RfC for chromium in the form of dissolved aerosols and mists (0.008 μg/m^3^) was used, the average concentration of chromium would result in an HQ exceeding 1.0.

The HQ for lead exceeds 1.0 at the high end of the range for CNS effects using CalEPA or USEPA reference values [32,33]. Lead is a known developmental neurotoxicant, and although no safe level exists, the MADL and NAAQS values allow for the inclusion of lead in the overall estimation of non-cancer adverse neurological and developmental effects.

At the low end of the reported ranges, no adverse non-cancer health risks were identified.

## 4. Discussion

This risk assessment found that some e-cigarettes may be significant sources of metal exposures, capable of increasing the risk of cancer and non-cancer respiratory and CNS health effects. Chromium and nickel, the key components of nichrome alloys, commonly used in heating elements, appear to be the main contributors to the cancer and non-cancer risks from vaping.

The calculated cancer risks from metals far exceed published exposures and risks to carcinogenic volatile organics such as formaldehyde or acetaldehyde from the same devices [35]. While not quantified here, the total cancer risk would be even higher when factoring in the contribution of carcinogenic volatile organics.

The published ranges of metal concentrations in e-liquids illustrate that chromium and nickel, at the average and high end of reported levels, occur in higher concentrations in e-cigarette liquids and aerosols than in tobacco smoke (discussed later) [36]. It is notable that even based on conservative assumptions, including using the lowest reported concentration for chromium, the estimated cancer risks are not negligible (5.7 × 10^−6^). Because there are no published data on the valence state or particulate nature of chromium in e-cigarette liquids, the risk assessment for chromium was based on the assumption that 100 percent of the chromium was in the VI+ form. This assumption maximizes the contribution to cancer risk from chromium; however, the overall sum of cancer risk was driven by the high concentrations of nickel.

In addition to cancer risk concerns, the potential non-cancer health risks from numerous metals, particularly on the respiratory tract are significant. The non-cancer risks are predominantly from nickel, with contributions from chromium, and manganese. The potential for the formation or aggravation of allergies or other atopic disease from e-cigarette exposures to nickel has not been explored.

To provide additional public health context about the metal composition data from our risk assessment, key aspects of the eight unique studies, including a review article, are summarized below.

Hess and colleagues reported a large range in metal concentrations in five nationally popular brands of cig-a-like type e-cigarettes [15]. In their study, metals, including cadmium, chromium, lead, manganese, and nickel were found in all of the liquids analyzed, with significant variation by brand. Individual metals were found to vary by up to 400-fold. The finding of high concentrations of nickel are of interest since nickel is a known human respiratory carcinogen and a known allergen (26,28). Because nickel is one element that occurs in significantly higher concentrations in e-cigarettes compared with conventional tobacco smoke, it represents a qualitatively new inhalation exposure and risk. Hess and colleagues described research in Japan and the USA that found nichrome and kanthal (an iron/chromium/aluminum alloy) in the heating coils in cig-a-likes. These researchers suggested that the metal contaminants could be addressed through increased product regulation that would cover all relevant metals [15].

Olmedo and colleagues reported metal concentrations in 56 e-liquid formulations in the context of the e-liquid refill dispensers (i.e., no contact with metal coils), compared to the same e-liquids in generated aerosols, and tanks [16]. In these studies, metal concentrations were universally higher in aerosols and the tanks than in the original e-liquid dispensers, suggesting that the contact of the liquid with the coils and/or heating and aerosolizing is the source of elevated metals.

Metals used in heating filaments can vary widely. Williams and colleagues found significant differences in metal wire compositions between e-cigarette and e-hookah brands, with the 16 most common elements across products including silicon, calcium, sodium, copper, magnesium, tin, lead, zinc, boron, selenium, aluminum, iron, germanium, antimony, nickel, and strontium. Twelve elements were found in four to seven of the products, including chromium, manganese, and molybdenum. Mercury was not detected in any product, and cadmium and arsenic were only rarely found [17,18].

A study of open and closed-system e-cigarette devices found substantial effects on the range of metals in aerosols as a function of power setting and device type [22]. In this study, two sets of median values were calculated across power settings of open devices from 40 to 200 W. Chromium and nickel concentrations were found to vary substantially between 40 and 120 W. Closed-system devices generally had metal concentrations that were a magnitude lower.

Na and colleagues reported that while e-liquid itself was low in metals, with nickel, chromium and iron below detection limits when initially analyzed, continued exposure of the liquid to the device’s heating elements during use dramatically increased the zinc, nickel, chromium, iron, and lead in the liquid. This demonstrates the transfer of heavy metals into the liquid over time and with e-cigarette use [37].

Saffari and colleagues originally reported that some particulate metals in secondhand emissions, including nickel, chromium, silver, and zinc, were higher in aerosols from e-cigarettes than are found in tobacco smoke. In their study, other metals, including cadmium and lead, were substantially lower in e-cigarette emissions [21]. Visser and colleagues, on the other hand, reported nickel, chromium, and arsenic as below the level of quantitation in secondhand aerosols of e-cigarettes [38].

In a more recent review of 12 studies, Gaur and Agnihotri described the numerous pathways for a range of metals, including carcinogenic metals such as chromium and nickel, to be introduced into e-cigarette liquids [24]. The presence of metallic structures, including the atomizer, cartomizer, or clearomizer, resistance wires, wicking element, and battery, combined with cyclic temperature changes within the device, result in metals leaching into the liquid before and during aerosolization. These researchers reported large variations in metal concentrations, of several orders of magnitude, both within brands and across manufacturers.

In addition to analytical determinations of metals in liquids and aerosols, there is published evidence of increased exposures to certain metals, as determined by urinary or blood biomarkers in e-cigarette users, when compared with non-users [39,40]. Notably, the concentrations of lead, nickel, copper, and manganese were elevated in urine compared with non-smokers. Nickel levels were nearly twice the concentration found in smokers (chromium not studied). These studies suggest that the metal exposures from e-cigarettes are bioavailable and become a systemic toxicity consideration.

Williams and associates reported the occurrence of metal nanoparticles containing tin, nickel, and chromium in e-cigarette cartomizers [17]. Particles greater than 1 µm in size contained tin, silver, iron, nickel, aluminum, and silicates, while particles less than 100 nm in diameter contained nickel, chromium, and tin. Nine of the 11 metals studied were found in higher concentrations than has been measured in cigarette smoke [17]. Mikheev and colleagues similarly found that the metal particles generated in e-cigarettes occur in the nanoparticle size range (i.e., less than 2.5 µm) [23]. The reported presence of nanoparticles is significant, as particles in the ultrafine size range are more hazardous to the lung than larger particles due to their ready access to alveolar region and rapid absorption systemically. Nanoparticles of nickel elicit greater inflammatory lesions in lungs of mice and rats than equivalent exposures to larger nickel particles [41,42]

The studies summarized above reveal that e-cigarettes are highly variable sources of toxicant exposures to users of these products. While a substantial and growing database of chemical composition of e-liquids and aerosols now exists, some key aspects remain obscure. Many people who use e-cigarettes do so with an implicit presumption, through marketing messages and some published study conclusions, that these devices provide nicotine with less exposure to toxic chemicals than with smoking cigarettes [29,43]. While it is true that e-cigarettes expose their user to markedly lower levels of numerous toxicants, including polycyclic aromatic hydrocarbons, many volatiles, and nitrosamines, it is apparent that exposures to some metals, including chromium, nickel, cadmium, manganese, and lead, are likely higher in e-cigarettes. The metals in the aerosols are believed to result from liquid contact with heated metallic components, or possibly as contaminants of nicotine extraction or other processes in the generation of the e-liquids themselves. For example, the tobacco plant concentrates some soil metals, including polonium, cadmium, and lead, depending on the soil content of these metals where the plant is grown, which could contribute to metal concentrations in e-liquids [24]. However, the evidence suggests that the nichrome heating element wires are a major source of chromium and nickel.

## 5. Conclusions

We found that e-cigarette liquids and aerosols are a significant and highly variable source of metals, leading to unacceptably high cancer and non-cancer risks at the mid and high end of the reported ranges. Chromium and nickel are the leading contributors to these risks, with cadmium, lead, manganese and arsenic as minor contributors. The large range of metals within and across e-cigarette brands indicates the need for improvements in product design, enforced product safety regulations and manufacturing quality control. Implementation of such measures could reduce metal exposures in e-cigarette users. In addition, the chemical species of chromium and nickel should be determined, and the particle size distribution of the metal particles characterized.

## Figures and Tables

**Table 1 ijerph-17-02146-t001:** List of studies and ranges of metal concentrations used in risk assessment in e-cigarettes.

Study.	Metric	Metals (Range)	Comments
Aerosol Studies			
Williams 2013 [17]	Aerosol: μg/10 puffs	Cr (0.007)	Single brand/design. Cartomizer
		Ni (0.005)	
		Mn (0.002)	
		Pb (0.017)	
		Cu (0.002)	
		Zn (0.058)	
		Sn (0.037)	
		Al (0.394)	
		Fe (0.52)	
		Ba (0.012)	
Goniewicz 2014 [19]	Aerosol: μg/10 puffs	Ni (0.007–0.019)	12 e-cigarettes from Poland and the UK
		Cd (0.001–0.015)	
		Pb (0.002–0.038)	
Mikheev 2016 [23]	Aerosol: ng/mg TPM	Cr (1.0–3.0)	Blu brand, 7 flavors
		Ni (0.1–0.2)	
		Pb nd	
		Cd nd	
		Mn nm	
		Cu (0.6–0.7)	
		Zn (2.0–3.0)	
		Sn (0.08–0.09)	
		Sb (0.1–0.4)	
		As (0.07–0.09)	
Gaur 2019 [24]	Aerosol: μg/10 puffs	Cr (0.007–0.2)	Review of 12 studies
		Ni (0.005–0.007)	
		Mn (0.002–0.035)	
		Pb (0.002–0.038)	
		Cd (0.0004–0.0146)	
		Cu (0.011–2.2)	
		Sn (0.036–6.0)	
		Al (0.266–0.394)	
**Liquid Studies**			
Hess 2017 [15]	Liquid: μg/L	Cr (56–726)	Cigalike refill liquids, N = 5
		Ni (58–15400)	
		Mn (26–918)	
		Pb (4.98–1630)	
		Cd (0.2–12.4)	
Olmedo 2018 [16]	Liquid: μg/L	Cr (55.4)	Multiple types of device. N = 56 aerosols and dispensers, N = 49 tanks
		Ni (233)	
		Mn (31.9)	
		Pb (40.2)	
		Cu (148)	
		Zn (426)	
		Sn (20.3)	
		Cd (0.126)	
		Sb (0.563)	
		Al (31.2)	
		Fe (382)	
Kamilari 2018 [20]	Liquid: μg/L	Cr (4–32)	Refill liquids, N = 22
		Ni (2–92)	
		Pb (1–11)	
		Cd (4.8–175)	
		As (nd–4)	
Zhao 2019 [22]	Liquid (from aerosol): μg/L	Cr (0.39–15.6)	Range of medians of 2 closed and 2 open devices
		Ni (1.3–2148)	
		Mn (0.39–64)	
		Pb (0.9–541)	
		Cd (0.04–0.16)	
		Cu (6.0–542)	
		Sb (0.15–4.6)	
		Sn (0.35–322)	
		Zn (683–3114)	
		Fe (3.4–153)	
		Al (4.1–17.7)	

**Table 2 ijerph-17-02146-t002:** Cancer potency slope factors and non-cancer reference exposure levels used for risk assessment of metals in e-cigarette liquids.

Cancer Potency (μg/kg/day)^−1^		Unit Risk (μg/m^3^)^−1^	Non-Cancer REL (μg/m^3^)	Target System
Cr	0.51^a^	0.15^a^ 0.012^b^	0.2^a^ 0.008^b^ (soluble) 0.1^b^ (particulate)	Respiratory system
Ni (subsulfide)	0.00091^a^	0.00026^a^ 0.00048^b^	0.014^a^	Respiratory, immunologic systems
Pb	0.000042^a^	0.000015^a^	0.5 (μg/d) (MADL)^a^ 0.15 (AAQS)^b^	CNS, reproductive, development
As	0.012^a^ 0.0015^b^	0.0033^a^ 0.0043^b^	0.015^a^	Development, cardiovascular, CNS, respiratory
Cd	0.015^a^	0.0042^a^ 0.0018^b^	0.02 ^a^	Kidney
Mn	NA	NA	0.09^a^ 0.05^b^	CNS

NA = not applicable; “a” = California Environmental Protection Agency (CalEPA) value (27); “b” =USEPA value (26); MADL = Maximum Allowed Daily Limit; REL = reference exposure level from the CalEPA.

**Table 3 ijerph-17-02146-t003:** Summary of estimated metal intakes, cancer and non-cancer risks from published metal concentration ranges in e-cigarette liquids and/or aerosols using CalEPA RELs, MADLs and cancer slope factors.

Estimated intakes* (μ g/kg/day)		Cancer Risks			Non-Cancer Hazard Quotients		
	Range	Low	High	Ave	Low	High	Ave
Cr	1.1 × 10^−5^ to 6.0 × 10^−2^	5.68 × 10^−6^	**3.07 × 10^−2^****	**3.10 × 10^−3^**	1.95 × 10^−4^	**1.06 × 10^+0^**	1.06 × 10^−1^
Ni	3.7 × 10^−5^ to 6.5 × 10^−1^	3.38 × 10^−8^	**5.88 × 10^−4^**	**5.24 × 10^−5^**	9.29 × 10^−3^	**1.61 × 10^+2^**	**1.44 × 10^+1^**
Pb	1.4 × 10^−5^ to 5.0 × 10^−2^	5.71 × 10^−10^	1.96 × 10^−6^	2.87 × 10^−7^	1.90 × 10^−3a^	**6.54 × 10^+0^**	9.58 × 10^−1^
Cd	3.6 × 10^−6^ to 3.5 × 10^−4^	5.40 × 10^−8^	**5.12 × 10^−5^**	**1.23 × 10^−5^**	6.30 × 10^−4^	5.98 × 10^−1^	1.43 × 10^−1^
As	0 to 1.1 × 10^−4^	-	1.37 × 10^−6^	6.86 × 10^−7^	-	2.67 × 10^−2^	2.67 × 10^−2^
Mn	3.1 × 10^−5^ to 2.6 × 10^−2^	NA	NA	NA	1.21 × 10^−3^	**1.02 × 10^+0^**	1.84 × 10^−1^

* per 2 mL e-liquid; NA = not applicable; a = MADL used in non-cancer calculation. ** Bold = significant risk.

**Table 4 ijerph-17-02146-t004:** Summary of estimated metal intakes, cancer and non-cancer risks from published metal concentration ranges in e-cigarette liquids and/or aerosols using U.S. Environmental Protection Agency (USEPA) reference exposure concentrations (RfCs), National Ambient Air Quality Standard (NAAQS), and cancer slope factors.

Estimated Intakes* (μg/kg/day)		Cancer Risks			Non-Cancer Hazard Quotients		
	Range	Low	High	Ave	Low	High	Ave
Cr	1.1 × 10^−5^ to 6.0 × 10^−2^	4.68 × 10^−7^	**2.53 × 10^−3**^**	**2.55 × 10^−4^**	3.90 × 10^−4^	**2.11 × 10^+0^**	2.13 × 10^−1^
Ni	3.7 × 10^−5^ to 6.5 × 10^−1^	6.24 × 10^−8^	**1.08 × 10^−3^**	**9.67 × 10^−5^**	-	-	-
Pb	1.4 × 10^−5^ to 5.0 × 10^−2^	5.71 × 10^−10^	1.96 × 10^−6^	2.87 × 10^−7^	3.17 × 10^−4a^	**1.09 × 10^+0a^**	1.61 × 10^−1a^
Cd	3.6 × 10^−6^ to 3.5 × 10^−4^	2.27 × 10^−8^	**2.15 × 10^−5^**	5.15 × 10^−6^	7.20 × 10^−6^	6.83 × 10^−3^	1.64 × 10^−3^
As	0 to 1.1 × 10^−4^	-	1.72 × 10^−6^	8.60 × 10^−7^	-	3.81 × 10^−4^	1.90 × 10^−4^
Mn	3.1 × 10^−5^ to 2.6 × 10^−2^	NA	NA	NA	2.18 × 10^−3^	**1.84 × 10^+0^**	3.32 × 10^−1^

* per 2 mL e-liquid; NA = not applicable; a = NAAQS used in non-cancer calculation. ** Bold indicates significant risk.

**Table 5 ijerph-17-02146-t005:** Non-cancer HI (summed) for the target organs * from estimated exposure to metals in e-cigarette aerosols (CalEPA RELs and MADL).

Endpoint	Low	Average	High	Metals
Respiratory	9.5 × 10^−3^	14.5	162	Cr, Ni
CNS	3.0 × 10^−2^	1.2	7.6	Pb, Mn, As
Reproduction	2.9 × 10^−2^	9.8 × 10^−1^	6.6	Pb, As
Renal	6.3 × 10^−4^	1.4 × 10^−1^	6.0 × 10^−1^	Cd

* Target organs as described by CalEPA [28].

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
