# Peer review of "Cancer and Non-Cancer Risk Concerns from Metals in Electronic Cigarette Liquids and Aerosols"

_ijerph, 2020, doi:10.3390/ijerph17062146_

Round 1

Reviewer 1 Report

This MS underlines the role of several metals in e-liquids and e-aerosols on cancer and non-cancer risks. It is understandable and the subject is important due to the impact on health.

Comments:

Line 23 - … metal exposure…

Line36 - showed

Line 67 – Please rephrase the sentence to avoid the word “data” twice in the same sentence.

Line 79 – How can authors assure that valence state of chromium was VI+ state for all chromium measurements? Please justify.

Please include a colun with references on Table 2.

Line 168 – Please include a reference in the sentence : …since nickel is a known human respiratory carcinogen and a known allergen.

Line 180 – Metals instead of The metals

Line 201 – described

Line 211 - and nickel levels

Please consider the following papers:

Gray N et al., Analysis of Toxic Metals in Liquid from Electronic Cigarettes. Int J Environ Res Public Health. 2019 Nov 13;16(22).

Halstead M et al., Analysis of Toxic Metals in Electronic Cigarette Aerosols Using a Novel Trap Design. J Anal Toxicol. 2019 Oct 4. pii: bkz078.

Jain RB. Concentrations of cadmium, lead, and mercury in blood among US cigarettes, cigars, electronic cigarettes, and dual cigarette-e-cigarette users. Environ Pollut. 2019 Aug;251:970-974.

Author Response

Comments:

Line 23 - … metal exposure…

Response: Done

Line36 - showed

Response: Done

Line 67 – Please rephrase the sentence to avoid the word “data” twice in the same sentence.

Response: Done

Line 79 – How can authors assure that valence state of chromium was VI+ state for all chromium measurements? Please justify.

Response: We have no such assurance. We have assumed this in absence of empirical data for the purpose of a conservative risk assessment. We discuss the implications of this assumption in the Discussion.

Please include a colun with references on Table 2.

Response: We agree and have opted for footnote instead of another column to economize on space.

Line 168 – Please include a reference in the sentence : …since nickel is a known human respiratory carcinogen and a known allergen.

Response: Done

Line 180 – Metals instead of The metals

Response: Done

Line 201 – described

Response: Done

Line 211 - and nickel levels

Response: Done

Please consider the following papers:

Gray N et al., Analysis of Toxic Metals in Liquid from Electronic Cigarettes. Int J Environ Res Public Health. 2019 Nov 13;16(22).

We thank the reviewer for drawing our attention to these references. We have considered these and the Gray reference is indeed relevant to our analysis but was published after we submitted the manuscript.  The metal concentrations in Gray et al are consistent with our reported range (i.e. would not alter our range), but could influence the average value somewhat.  Additional studies will continue to be published as well.  We do not consider that the time spent adding the Gray et al., data into our manuscript is justified as the impact on our conclusions would be minimal.

The other 2 references are of interest, but do not contain data that would enter into our exposure or risk calculations. Notably our paper is consistent with these papers.

Reviewer 2 Report

Dear Authors, 

The manuscript titled "Cancer and Non-Cancer Risk Concerns from Metals in Electronic Cigarette Liquids and Aerosols" aims to review the deleterious effects of metals present in e-cigarettes.

The paper sounds good and it is well written and logically constructed.

However, in my opinion, the Authors need to increase the background studies in the introduction. For example, the should mention the study by Vesser et al., 2019 (The Health Risks of Electronic Cigarette Use to Bystanders) in which is highlighted that "an increased risk of cancer could not be excluded".

Furthermore, another interesting article (Jankowski et al., 2019: E-Cigarettes are More Addictive than Traditional Cigarettes—A Study in Highly Educated Young People), should be mentioned in the discussion/conclusion section, in order to underline the public health risk concerning the e-cigarette addiction.

Finally, read carefully the paper to avoid typos. Indeed, in the introduction section (page 2, lane 54), reference 21 is missed (it appears for the first time in the discussion section, page 7, lane 200).

Author Response

We thank the reviewer for their constructive comments. Responses are below:

Reviewer comments/responses

However, in my opinion, the Authors need to increase the background studies in the introduction. For example, the should mention the study by Vesser et al., 2019 (The Health Risks of Electronic Cigarette Use to Bystanders) in which is highlighted that "an increased risk of cancer could not be excluded".

Furthermore, another interesting article (Jankowski et al., 2019: E-Cigarettes are More Addictive than Traditional Cigarettes—A Study in Highly Educated Young People), should be mentioned in the discussion/conclusion section, in order to underline the public health risk concerning the e-cigarette addiction.

Finally, read carefully the paper to avoid typos. Indeed, in the introduction section (page 2, lane 54), reference 21 is missed (it appears for the first time in the discussion section, page 7, lane 200).

Response:

We have added the Visser et al., 2019 reference in the discussion as reference #38. We did not consider that inclusion in the introduction was the appropriate place to include this. The Jankowski et al., 2019 paper relates to addiction rather than metal exposures and risks, and we decline to include this in our paper.

We have addressed the mentioned typo on page 2.